# Preliminary Assessment of COVID-19 Implications for the Water and Sanitation Sector in Latin America and the Caribbean

**DOI:** 10.3390/ijerph182111703

**Published:** 2021-11-08

**Authors:** Miguel de França Doria, Patricia Segurado, Marcelo Korc, Leo Heller, Blanca Jimenez Cisneros, Paul R. Hunter, Martin Forde

**Affiliations:** 1Intergovernmental Hydrological Programme (IHP), UNESCO Regional Office, Montevideo 11200, Uruguay; 2Regional Water and Sanitation Technical Team (ETRAS), Pan American Health Organization, PAHO, Lima 15038, Peru; 3Climate Change and Environmental Determinants of Health Unit, Pan American Health Organization, PAHO/WHO, Washington, DC 20037, USA; korcmarc@paho.org; 4René Rachou Institute, Oswaldo Cruz Foundation, Belo Horizonte 30190-009, MG, Brazil; leo.heller@fiocruz.br; 5National Water Commission (CONAGUA), Mexico City 04340, Mexico; 6Health Protection Department, Norwich Medical School, University of East Anglia, Norwich NR4 7TJ, UK; paul.hunter@uea.ac.uk; 7Department of Public Health and Preventive Medicine, St. George’s University, St. George’s P.O. Box 7, West Indies, Grenada; MForde@sgu.edu

**Keywords:** water and sanitation, COVID-19 pandemic

## Abstract

COVID-19 has had a severe impact on human health, as well as in social and economic terms, with implications for the management and governance of the water and sanitation sector. These implications are evident in Latin America and the Caribbean due to existing challenges the region faces in accessing water and sanitation services. In spite of significant advances, around 65 million people in the LAC region currently lack appropriate access to water and soap to wash their hands—one of the most basic measures to prevent the spread of disease. Furthermore, social and economic vulnerabilities have exacerbated the effects of the pandemic in the region, particularly among those living in poverty. The COVID-19 pandemic thus requires the mobilization of frameworks such as the human rights to water and sanitation, specifically considering the region’s realities. This paper provides a review of some of the challenges currently faced in the region and advances a series of recommendations for enhancing access to water, sanitation and hygiene. The importance of effective governance, management and communication strategies in the water provisioning sector is highlighted in the context of the pandemic, and the role of science and research for adequate decision making is emphasized.

## 1. Introduction

This paper is based on the outcomes of strategic public meetings conducted since March 2020, with the objective of strengthening the response of the water sector to the COVID-19 pandemic in Latin America and the Caribbean.

The COVID-19 pandemic was caused by a coronavirus (SARS-CoV-2) that spread worldwide, with over 170 million confirmed infections and around 3.7 million confirmed deaths as of 7 June 2021 [1]. The infection spreads primarily from one person to another by contact and respiratory droplets that are expelled orally. A susceptible person may inhale these droplets from the air or, less frequently, the droplets may land on surfaces (such as tables and door handles) that people touch and subsequently transfer the virus to their faces [2]. For these reasons, the most important measures that can be taken in terms of prevention are to keep physical distance between people, adequately ventilate indoor spaces and wash hands with water and soap, which dissolves the lipid envelope of the virus. The World Health Organization (WHO) issued a technical brief for the prevention of COVID-19, targeted at water and sanitation practitioners and providers [3]. This document emphasizes the key importance of hand hygiene in preventing infection, and recommends that Water, Sanitation and Hygiene (WASH) practitioners work to enable more frequent hand hygiene with water and soap or using alcohol-based handrubs (ABHRs). It further states that the current WHO guidelines for the safe management of drinking water and sanitation services are sufficient in the context of the COVID-19 outbreak. It also highlights many co-benefits that can be realized by safely managing water and sanitation services and applying good hygiene practices.

The primary means by which the virus spreads is clearly through respiratory droplets and close contact, but some concerns were raised that the infection may also be spread fecal-orally. This is not impossible as during the Severe Acute Respiratory Syndrome (SARS) epidemic of 2002–2003 there was one cluster of cases associated with a defective wastewater system in an apartment block [4]. At the time, SARS-CoV was found in sewage wastewater and found to be active from 2 days at 20 °C to 14 days at 4 °C [5]. During the current pandemic, RNA from SARS-CoV-2 has been detected in the feces of patients with COVID-19, but usually it has not been possible to isolate the active virus, suggesting that the fecal route may not be infectious. While several studies have detected SARS-CoV-2 viral RNA fragments in the fecal matter of patients throughout their illness and after recovery, current evidence underscores the difficulty of culturing the virus in excreta. Three studies reported the infectious virus in feces, while others have not found infectious virus in this medium. One study found infectious SARS-CoV-2 in the urine of one patient and detected viral RNA in gastrointestinal tissue [6]. Preliminary research has found SARS-CoV-2 genetic material in raw sewage but not in treated effluents, and wastewater analyses have been used to determine COVID-19 prevalence among some populations [3]. In addition, enveloped viruses such as SARS-CoV-2 tend to be quite sensitive to halogen disinfectants and ultraviolet disinfection [2]. For these reasons, treated drinking water is unlikely to pose a significant risk of COVID-19 and wastewater is also unlikely to be an important risk factor, particularly in high- and middle-income areas where adequate water and wastewater systems are in place. Although fecal transmission cannot be entirely ruled out, such transmission—if it exists—would likely only play a minor role in the pandemic. A number of co-incident factors need to occur to increase the likelihood of water outbreaks [7]. Outbreaks with lipid-layered viruses are uncommon and the water pathway will only be potentially viable if: (1) the presence of active virus in stool is verified (2) in areas with a large number and high density of COVID-19 cases (e.g., hospitals and some urban areas) (3) with an inadequate sanitation or sewage system, presenting drainage failures inside buildings (as was the case reported during the 2002–2003 SARS outbreak) or (4) that contaminate a nearby deficient drinking water distribution system with contaminant ingress in the absence of disinfectants (5) in a period of time that is sufficiently rapid for the virus to remain active and (6) with a lack of monitoring to prompt rapid corrective measures. The contamination of drinking water systems via their sources is even more unlikely due to the relatively long time necessary for the full cycle, and the fact that lipid-layered viruses in water tend to adhere to particles that are usually largely removed in drinking water treatment systems, with disinfection deactivating any remaining viruses. The contamination of recreational waters and of urban runoff water that comes into contact with a population may present additional challenges.

There are other reasons for maintaining, reinforcing or implementing safe water and sanitation services in order to minimize the occurrence of other infectious diseases. Comorbidity may be a risk factor for COVID-19 outcomes, and in several countries, health institutions were overloaded at different stages of the pandemic [8]. Eliminating other causes of hospitalization would alleviate these institutions and allow them to focus on COVID-19. Socioeconomic status may also play a crucial role in the pandemic. People who live in poverty, in informal and migrant settlements, are homeless or are incarcerated in prisons are much more likely to be affected for several reasons, ranging from a lack of information and resources to protect themselves to the adverse living conditions in crowded settlements and a lack of access to sufficient water for handwashing. Thus, the COVID-19 pandemic requires a mobilization of frameworks that specifically focus on these groups, such as the framework on the human rights to water and sanitation [9].

This paper emerged from public discussions held in April 2020 onward by different key actors on open web-based seminars—organized by UNESCO’s Intergovernmental Hydrological Programme, WHO Americas and other partners—to discuss the potential implications of the pandemic for the water sector. This paper expands on those discussions with the purpose of exploring some of the main interlinkages between the water sector and COVID-19, and exploring recommendations to improve responses at different levels.

## 2. Context in Latin America and the Caribbean

Latin America and the Caribbean (LAC) are home to over 650 million people. As of 15 April 2020 (50 days after the first case was reported in LAC), over 70,000 people were confirmed as infected by SARS-CoV-2 across almost the entire region. Most LAC countries have adopted physical distancing measures to prevent the dissemination of the coronavirus through person-to-person contact, suspending or restricting the function of educational and cultural institutions as well as that of noncritical business. In this relatively short period, COVID-19 has had a very significant social and economic impact, the real extent of which is yet to be determined, notably on those from vulnerable groups. In addition, many migrants are returning to their country or area of origin, potentially disseminating the virus among particularly vulnerable communities [10]. For low-income and middle-income countries and deprived areas that have very limited resources to address pandemics, this influx can place additional pressure on water services that are already facing considerable difficulties.

As previously stated, a key recommendation given by health authorities to help mitigate the spread of SARS-CoV-2 is to frequently wash hands with water and soap [11]. The estimates for the 12 LAC countries for which hygiene data is available (representing about 40% of the region population) indicate that 24% (over 65 million people) lack appropriate access to hygiene [12]. Of this number, about 12% have limited hygiene (without water or soap) and another 12% completely lack hygiene facilities. This situation is worse in rural areas, where 22% have limited access to hygiene and 17% have no access to hygiene. Thus, access to hygiene is an issue of prime concern, particularly among vulnerable populations.

For the water sector, the current situation with SARS-CoV-2 may be further exacerbated by existing deficiencies of the drinking water supply, urban wastewater treatment systems and of other water-related pathways for microbiological contamination. Latin America and the Caribbean is the most urbanized region on the planet, with over 80% of the population living in cities, including 20% in slums, and its high human density can contribute to the propagation of pathogens—particularly airborne and waterborne viruses [13]. The region has six megacities with over ten million inhabitants each, and a large number of big cities contain peri-urban areas, slums and informal settlements that are often extremely crowded and face a number of challenges in terms of water supply and sanitation [14]. While around 87% of LAC’s population has access to sanitation, only 31% have access to some form of safely managed services [8]. Around 2.2% of LAC’s population (nearly 15 million people) still practice open defecation. Water access is generally higher in urban areas (97% access) and very significant progress was achieved over the last few decades, but several urban water supply systems face problems with effective disinfection, pressure, supply continuity and water quality [15]. In rural areas, around 15% of the population still lack access to a water supply. Episodic contamination of drinking water sources by fecal microorganisms from human and non-human sources is a challenge [12].

In addition, Latin America and the Caribbean are characterized by very significant hydrological, social, economic and environmental diversity, which often manifest within and across countries. As an example, the region includes the world’s largest river basin (Amazon) and two of the largest aquifers (Amazon and Guarani), but also the driest non-polar desert in the planet (Atacama). While in some respects this diversity can be considered one of the region’s strengths (e.g., by contributing to its resilience), it also complicates generalizations and water resource management with inequalities presenting considerable challenges.

A key issue of concern is the inadequate financing for the maintenance of the existing systems and the urgent need for water provisioning systems to be significantly upgraded and improved, especially in areas where the most vulnerable segments of the population are located, such as rural populations and native peoples. Water-quality monitoring systems are often limited or nonexistent due to tight financial resources. In addition, many LAC water systems do not have the ability to maintain adequate disinfection levels throughout the whole distribution network, with peri-urban areas located in the fringes of the network suffering increased risks [16]. If the current efforts to constrain and address the risks posed by COVID-19 (including its social and financial impacts) leave the water and sanitation sector behind, the rapid deterioration of existing water provisioning systems will likely pose significant risks to human health in the short and long term [12]. Affordability and inequality issues are particularly noteworthy. In some countries, water services are intermittent due to the high cost of electricity, which can represent around 65% of the operating costs. The high energy cost is also one of the reasons why wastewater treatment plants are not in operation, even when infrastructure is in place. The benefits of investing in water and sanitation largely outweigh the costs. An investment of USD 1 in water and sanitation services has a return in the range from USD 5 to USD 28, also encompassing reductions in health care expenditures [17]. Furthermore, the H1N1 outbreak in Mexico showed that if investments were made to the school water services, the rate and number of days where economic activities were stopped could have been significantly reduced. Financing for scientific research is also scarce in LAC, and this is reflected in a significantly lower number of researchers and research outputs when compared with other regions [18]. As knowledge and innovation are key to advancing policies and informed decision making and management, limitations in this field also limit the adoption of adequate and efficient solutions, with particular relevance to the local level.

Overall, these needs and priorities are addressed by Agenda 2030, in particular its Sustainable Development Goals (SDGs), whose implementation becomes even more urgent in the context of COVID-19. Among other objectives, the SDGs aim at: (1) achieving universal and equitable access to safe and affordable drinking water for all; (2) achieving access to adequate and equitable sanitation and hygiene for all and ending open defecation, paying special attention to the needs of women, girls and those in vulnerable situations; and (3) reducing the proportion of untreated wastewater.

## 3. Preliminary Recommendations for the Water Sector

The foremost recommendation for decision makers and managers from the water sector is to closely follow the guidelines and instructions of their national health services and coordinate with them in the context of COVID-19. A specific summary of such recommendations and guidelines has been developed in the context of COVID-19 and should be disseminated among water and sanitation service providers [2]. The preliminary recommendations presented in this paper regarding COVID-19 emphasize the importance of implementing the relevant international recommendations, good practices and existing regulations. The facilitation of the flow of information between key stakeholders, coupled with the establishment of multiple protective barriers throughout the water supply and wastewater systems, are necessary components of a successful strategy to reduce the risk of contamination.

The strongest contribution of the water sector to pandemic containment would be to place an absolute priority on universal access to water for hygiene, particularly handwashing. Therefore, the following preliminary recommendations have been derived:Urgently advance efforts to provide universal access to safe water in sufficient quantities for hygiene and vital human needs. In the context of the pandemic, focus should be on access to hygiene for populations living in the most vulnerable conditions. This includes the provision of available, accessible and affordable water for the homeless, detention facilities, low-income elderly peoples’ nursing homes and particularly informal settlements.In areas under water stress or scarcity, or with water supply deficiencies, consider the provision of ABHRs. One cubic meter of water is enough to wash hands 250 to 300 times, while a liter of gel allows up to 1000 applications. In addition, coordinate with Health Authorities to ensure that recommended hygiene practices are appropriately communicated and understood. In such areas, these messages may be accompanied by messages emphasizing the need to use water wisely. Consider the need for water for additional hygiene practices, such as to shower and to wash exposed clothing. Provisions must be made for the dry season and priorities in allocations may be needed. Water rationing schemes (e.g., by forbidding the irrigation of gardens) must be required and implemented to allow wider access for priorities.A short-term action is to place hand hygiene stations at the entrance of public buildings (including schools and health centers), private commercial buildings and other strategic sites of high public affluence, such as transportation locations (especially major bus and train stations, airports and ports). The quantity and ease of use of hand hygiene stations should be tailored to the population type (e.g., children, elderly, those with limited mobility).

Concerning drinking water and wastewater services, in addition to hygiene, the main recommendations are:Ensure the supply of safe water to hospitals and other healthcare facilities, including those recently built or in converted buildings, using water tanks containing disinfected water if necessary [19]. Ensure that these facilities have adequate sewage systems.Isolation and frequent hygiene habits are leading to an increase in water consumption (e.g., currently around 20% in parts of Mexico), changing the water demand geography and demand curve. As some immediate operational adjustments are impossible, water tank trucks may be used to address need.The deterioration of the financial and economic environment due to the COVID-19 pandemic may have serious impacts on the water sector. Unpaid bills at a period when credit is also difficult to obtain should not disrupt maintenance or the acquisition of disinfection materials. Financial mechanisms should be put in place so as to ensure that water services are maintained and that the water supply is not discontinued to those that cannot pay their bills. Contingency plans should include economic considerations, which should be communicated to relevant national authorities. At the policy level, it is essential to revert disconnections to water services due to an inability to pay bills. Mechanisms should be considered for waiving payments of those in poverty. It is key to coordinate with power companies for similar approaches due to their relevance for water supply.

Secondary recommendations include:Keep water supplies safe with disinfection. There must be a residual concentration of free chlorine of ≥0.5 mg/L at least 30 min after coming into contact with water, with a pH < 8.0. Ensure monitoring for the presence of adequate levels of chorine or other disinfectants along the distribution system, particularly in systems with a history of a long-standing lack of residual chlorine or recent fecal contamination and in systems that supply large numbers of people. Perform a rapid assessment of the water disinfection facilities. Ensure that there are strategic stocks of reagents and materials in case of the disruption of deliveries.Adopt a multi-barrier approach to focus and reduce the risk of waterborne outbreaks, promoting the water safety plans (WSPs) proposed in the WHO Guidelines for drinking water quality [5]. This involves identifying and controlling potential drinking water contamination both at the source and in the distribution system.Develop plans for temporary replacements of staff of water treatment and sewage plants that may become ill, particularly for those in critical operational units. Ensure that staff can access their workplace during eventual disruptions of public transportation. Ensure that staff exposed to sewage have appropriate personal protective equipment and adhere to safety and hygiene procedures.

Ensure proper governance, management and communication strategies to:Allow for the easy exchange of information and coordination with national health authorities and with those in charge of social, rural, food and agricultural portfolios. Report critical issues of concern and inform about other potential pathways via untreated sewage (including potentially contaminated recreational waters and areas where sewage is used for the production of vegetables), considering the different risks—especially microbiological—due to the lack of wastewater treatment.Communicate risks and build trust with the general population, taking into account local risk perception processes. Keep in mind that many communities speak native languages and there are cultural aspects involved; therefore, materials should be adapted and/or schematic figures used, considering local values and beliefs.Develop and disseminate guideline materials to assist rural communities in disinfecting their drinking water supplies. Distribute chlorine tablets, silver or iodine disinfectants to vulnerable communities to allow the local or at-home treatment of drinking water supplies [20].Advocate for adequate financial mechanisms to ensure universal access to water, sanitation and hygiene, and allow the sustainability of these systems within the context of the SDGs, fostering cooperation and coordination at all levels with the participation of communities.Emphasize epidemiological considerations in policies and guidelines for institutions in charge of water, in particular of drinking water and sanitation services, to strengthen the actions associated with the sanitary surveillance of drinking water and the sanitation chain (the collection, treatment, reuse and/or disposal of human waste in formal and informal settings). Develop mechanisms to reduce high levels of contamination in water sources and water bodies, to protect small basins, to strengthen the coverage of wastewater treatment, and to foster the elaboration of sanitation security plans as responses for the post-COVID-19 period.

Scientific research is fundamental for evidence-based decision-making. At the time of preparation of these recommendations, research on potential interlinkages between COVID-19 and water and sanitation was limited. Future research thus should:Identify additional opportunities for and barriers to the implementation of water, sanitation and hygiene measures. Assess deficiencies in existing water and sanitation systems that may increase the risk of propagating SARS-CoV-2 or comorbidity. Pursue efforts to develop and implement low-cost water and sanitation services, particularly for vulnerable populations that currently lack such services. Assess the extent to which SARS-CoV-2 makes an even stronger case for investing in WASH and how it differs from what is currently required. Assess the potential role of water and sewage in the transmission routes of SARS-CoV-2 in areas where provision is precarious, untreated water is provided, and there is intermittent supply and unsafe sanitation. Several studies have demonstrated that increases in SARS-CoV-2 RNA can be detected in environmental samples several days before the detection of COVID-19 through clinical surveillance. Consequently, there is the potential to use environmental surveillance for early warning, particularly of clusters or outbreaks in countries that have contained transmission and are easing public health and social measures or in the event of seasonality [6].Develop scenarios, forecasts and tools to support decision-making during water-related emergencies, such as floods and droughts, which may coincide with the pandemic.Further research the presence of SARS-CoV-2 in sewage and its relevance to COVID-19, as long as this supports and does not compromise the capacity of national health authorities to address the current pandemic due to shortages of reagents and microbiologists [21]. Explore the potential of sewage samples for the identification of virus variants and for estimates of prevalence.Cross-basin, cross-national and cross-regional comparisons can help to further identify geographical specificities and determine the full extent of potential direct and indirect relationships between water management and the current pandemic. Such studies should also attempt to capture the significant environmental, social and economic diversity of LAC in order to better understand underlying factors mediating or contributing to other findings.

## 4. Conclusions

In the context of providing water services in the midst of the COVID-19 pandemic, a key priority is to rapidly ensure easy and universal access to this vital resource for hygiene for all segments of the population, with special attention given to vulnerable groups. Additionally, in order to promote the inactivation of the SARS-CoV-2 virus, proper attention needs to be given to the management, operation and maintenance of water and sanitation systems. Where deficiencies are identified, they must be promptly dealt with so as to ensure that any propagation of COVID-19 through possible water pathways is, if not eliminated, greatly minimized [22].

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
