# Peer review of "Preliminary Assessment of COVID-19 Implications for the Water and Sanitation Sector in Latin America and the Caribbean"

_ijerph, 2021, doi:10.3390/ijerph182111703_

Round 1
Reviewer 1 Report
The paper is focused on the very important subject of COVID-19 implications for the water and sanitation sector. Some changes to the article are necessary. I present my comments to the work below:
- There is no clear indication of the purpose of the work in the Introduction section. At the end of the Introduction, a summary of the current state of knowledge and the purpose of the article should be presented.
- In the article, I miss a separate chapter with the discussion of the presented content. The last points in chapter 3 - about the need for research, should already be included in the discussion and it would be good to expand this content. Additionally, the discussion should indicate potential problems, difficulties with introducing the changes indicated in point 3 of the paper.
- The topic of COVID-19 is of course a relatively new subject in the literature. However, it would be good to expand the scope of the references. In Introduction section, you list the conditions that must exist for water pathway to be potentially viable, but you do not give the source. Is it from 5. reference (Risebro et al.) or from another source?
- In Introduction section, you write: "„Preliminary research has found SARS-CoV-2 genetic material in raw sewage but not on treated effluents.2”. What is „2” – annotation or reference?
- You mention about megacities and areas with very high-density population. It would be good if you could add some map with Latin America and the Caribbean including this information.
Author Response
The authors are grateful to the reviewer for these comments. We feel the comments helped to improve the paper and to include a few aspects that were not considered before. The way the comments were integrated into the manuscript is detailed in the attached file, in yellow for ease of identification. The changes are also visible in the manuscript, in track changes.

Reviewer 2 Report
Overall comments:
The paper could benefit from providing a stronger case (or not) for the extent to which SARS-COV2 makes an even stronger case for investing in WASH, and how it differs from what is already required. What is different about the recommendations because of SARS-COV2 and what is just the same as before? It could also make a stronger case for what is particularly different (or not) about LAC and central America compared to other parts of the world. As an example, piped water supply prevalence is much greater than in Africa – what impact does that have.
The paper generalises across the region. Are there no differences between the countries, and is there any learning from one country that could be important for another? Some visual data on water, sanitation and in particular hygiene access across the country could really lift the paper – particularly if placed alongside estimates of Covid infections (noting that these will be revised in the future anyway).
The last four conclusions are perhaps the most interesting in relation to Covid, but to whom are the recommendations addressed to? As they stand, they are quite general, and not directed, so not as useful for policy-making as they could be.
Detailed comments:
PP1: “COVID-19 caused by a coronavirus (SARS-CoV-2) has spread worldwide during 2020, with over 122.536.880 people infected and around 2.703.780 deaths.[1]” – please indicate clearly the time period and note that this is an estimation.
“A susceptible person may then inhale these droplets from the air or the droplets may land on surfaces such as tables and door handles, that people touch and then transfer the virus to their faces. – include recent research on the relative importance of surfaces relative to airborne. Add the importance of ventilation.
Worth mentioning that mentoring of wastewater has been a way of measuring prevalence (e.g. in Switzerland
. Comorbidity is a risk factor for COVID-19 outcomes and in several countries health institutions are overloaded during the pandemic. Include some of these countries and references, with data.
In addition, many migrants are returning their country or area of origin, potentially disseminating the virus among particularly vulnerable communities Include references.
Latin America and the Caribbean is the most urbanized region in the planet, with over 80% of the population living in cities, including 20% in slums, Include references.
Water access is generally higher in urban areas (97% access), but many urban water supply systems face problems of effective disinfection, pressure, supply continuity and water quality. In rural areas around 15% of the population still lack access to water supply. Include references.
Episodic contamination of drinking water sources by fecal microorganisms from human and non-human sources is a challenge. Include references.
A key issue of concern is the inadequate financing for the maintenance of the existing systems and the urgent need for water provisioning systems to be significantly upgraded and improved, including in areas where the most vulnerable segments of the population are located. Water quality monitoring systems are often limited or non-existent due to tight financial resources. In addition, many LAC water systems do not have the ability to maintain adequate disinfection levels throughout the whole distribution network, with peri-urban areas located in the fringes of the network suffering from increased risks. If the current efforts to constrain and address the risks posed by COVID-19, including its social and financial impacts, leave the water and sanitation sector behind, the rapid deterioration of existing water provisioning systems will pose significant risks to human health. Affordability issues and inequalities are particularly noteworthy. In some countries water services are intermittent due to the high cost of electricity, which can represent around 65% of the operating costs. The high energy cost is also one of the reasons why wastewater treatment plants do not operate, event when infrastructure is in place. Include references throughout.
Suggest grouping the recomendations into categories that make them easier to digest. 19 are a lot.
Author Response

(The authors gave the same response as above.)

Round 2
Reviewer 1 Report
Many thanks to the authors for their response to all of my comments. I found the article improved. All my suggestions were considered and the most important changes were made.
I agree with the authors that it would be a good idea to check the format of references in the final version of the paper.
I do not have any other comments and highly value the article for the relevance of the subject.
Reviewer 2 Report
Well done with the improvements. A nice paper.